

# First fossil species of family Hyidae (Arachnida: Pseudoscorpiones) confirms 99 million years of ecological stasis in a Gondwanan lineage

Liza M. Röschmann[1,2], Mark S. Harvey[3], Yanmeng Hou[4], Danilo Harms[2], Ulrich Kotthoff[5], Jörg U. Hammel[6], Dong Ren[4] and Stephanie F. Loria[2]

[1] Department of Geology, Universität Hamburg, Hamburg, Germany
[2] Section Arachnology, Centre for Taxonomy and Morphology, Museum of Nature Hamburg—Zoology, Leibniz Institute for the Analysis of Biodiversity Change, Hamburg, Germany
[3] Collections & Research, Western Australian Museum, Welshpool, Australia
[4] College of Life Sciences, Capital Normal University, Beijing, China
[5] Centre for Biomonitoring and Conservation Science, Museum of Nature Hamburg—Geology, Leibniz Institute for the Analysis of Biodiversity Change, Hamburg, Germany
[6] Institute of Materials Physics, Helmholtz-Zentrum Hereon, Geesthacht, Germany

Corresponding author
Stephanie F. Loria,
s.loria@leibniz-lib.de

## ABSTRACT

Burmese amber preserves a diverse assemblage of Cretaceous arachnids, and among pseudoscorpions (Arachnida: Pseudoscorpiones), ten species in five families have already been named. Here, we describe a new fossil species from Burmese amber in the pseudoscorpion family Hyidae, providing detailed measurements, photographs and 3D-models from synchrotron scanning. Based on morphology, the new fossil, *Hya fynni* sp. nov. is placed in the genus *Hya*, and is nearly identical to extant species in the genus, except for the position of trichobothrium *est* on the pedipalpal chela, thereby indicating extreme morphological stasis in this invertebrate lineage over the last 99 million years. *Hya fynni* represents the first described fossil species in Hyidae, and the third described Burmese fossil in the superfamily Neobisioidea. It also joins the garypinid, *Amblyolpium burmiticum*, in representing the oldest fossil records for extant pseudoscorpion genera. Considering proposed divergence dates, the newly described fossil species bolsters a Gondwanan origin for Hyidae, and provides evidence for the "*Late Jurassic Rifting*" hypothesis for the Burma Terrane, in which this landmass rifted from Gondwana in the Late Jurassic and collided with Eurasia by the Cretaceous/Eocene. Like *Hya* species today, *H. fynni* likely inhabited humicolous microhabitats in tropical forests on the Burma Terrane, supporting ecological niche stasis for this family since the Mesozoic.

## INTRODUCTION

Burmese amber, also called burmite is found in the Hukawng Valley in northern Myanmar (Fig. 1). It is arguably one of the most important Cretaceous deposits, preserving a rich
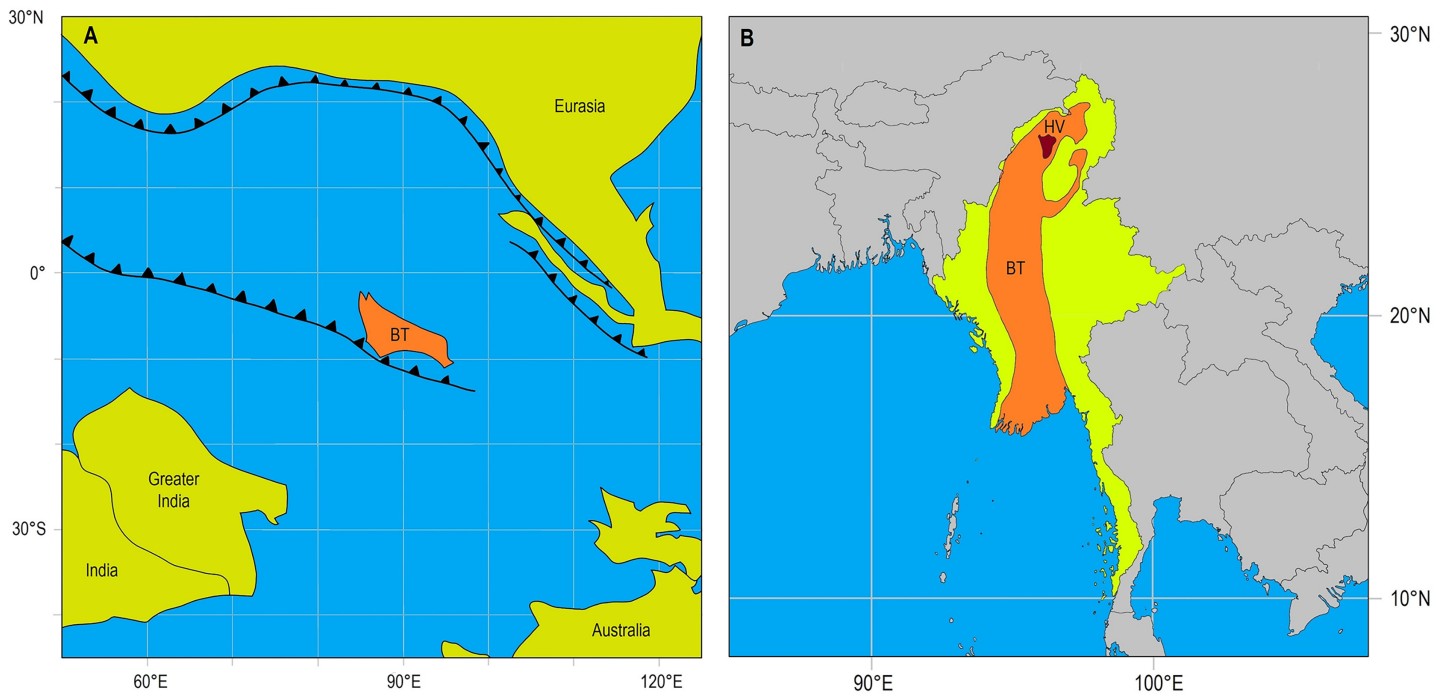

**Figure 1 Location of Burma Terrane in the Late Cretaceous and today.** (A) Position of the Burma Terrane (BT) during the time of Burmese amber formation in the Late Cretaceous (~95 Ma). (B) Location of Hukawng Valley (HV) on the Burma Terrane (BT) in Myanmar today. Reconstructions adapted from *Westerweel et al. (2019)*.

biota of arthropods, gymnosperms, angiosperms, vertebrates, fungi and mollusks (*Grimaldi, Engel & Nascimbene, 2002*; *Selden & Ren, 2017*). Burmese amber likely originates from the resin of conifer trees in the family Araucariaceae (*Poinar, 2007*, *2018*), and recent age estimates date it to the Early Cenomanian (98.79 ± 0.62 Ma) (*Shi et al., 2012*) in the Late Cretaceous. At this time, the Burma Terrane (BT), the landblock where Burmese amber is located, had already rifted from Gondwana, either in the Devonian (*Metcalfe, 2011*, *2013*) or Jurassic (*Seton et al., 2012*; *Westerweel et al., 2019*). However, whether the BT had already collided with Eurasia at the time of amber formation is debated (*Metcalfe, 2011*, *2013*; *Seton et al., 2012*; *Westerweel et al., 2019*). The climate was also much warmer, and the Burma Terrane is believed to have supported a tropical rainforest (*Poinar, 2018*; *Xing et al., 2018*) with caves and lacustrine environments.

Pseudoscorpiones are a mesodiverse order of arachnids and because of their soft tissue, and small size (ca. 2–12 mm in body length), they rarely fossilize in rock (*Harms & Dunlop, 2017*; *Johnson et al., 2023*). Besides the oldest known pseudoscorpion, *Dracochela deprehendor Schawaller, Shear & Bonamo, 1991* in the extinct family Dracochelidae *Schawaller, Shear & Bonamo, 1991* from Mid-Devonian (~380–374 Ma) deposits in Gilboa, New York (*Schawaller, Shear & Bonamo, 1991*; *Judson, 2012*), and the oldest known extant family record, *Archaeofeaella hendericxi Kolesnikov et al., 2022* in Feaellidae Ellingsen, 1906 from Upper Triassic deposits in Ukraine (*Kolesnikov et al., 2022*), all other pseudoscorpion fossils are found in Mesozoic or Cenozoic ambers (Fig. 2) (*Cockerell, 1917*, *1920*; *Schawaller, Shear & Bonamo, 1991*; *Judson, 2009*, *2012*, *2017*;

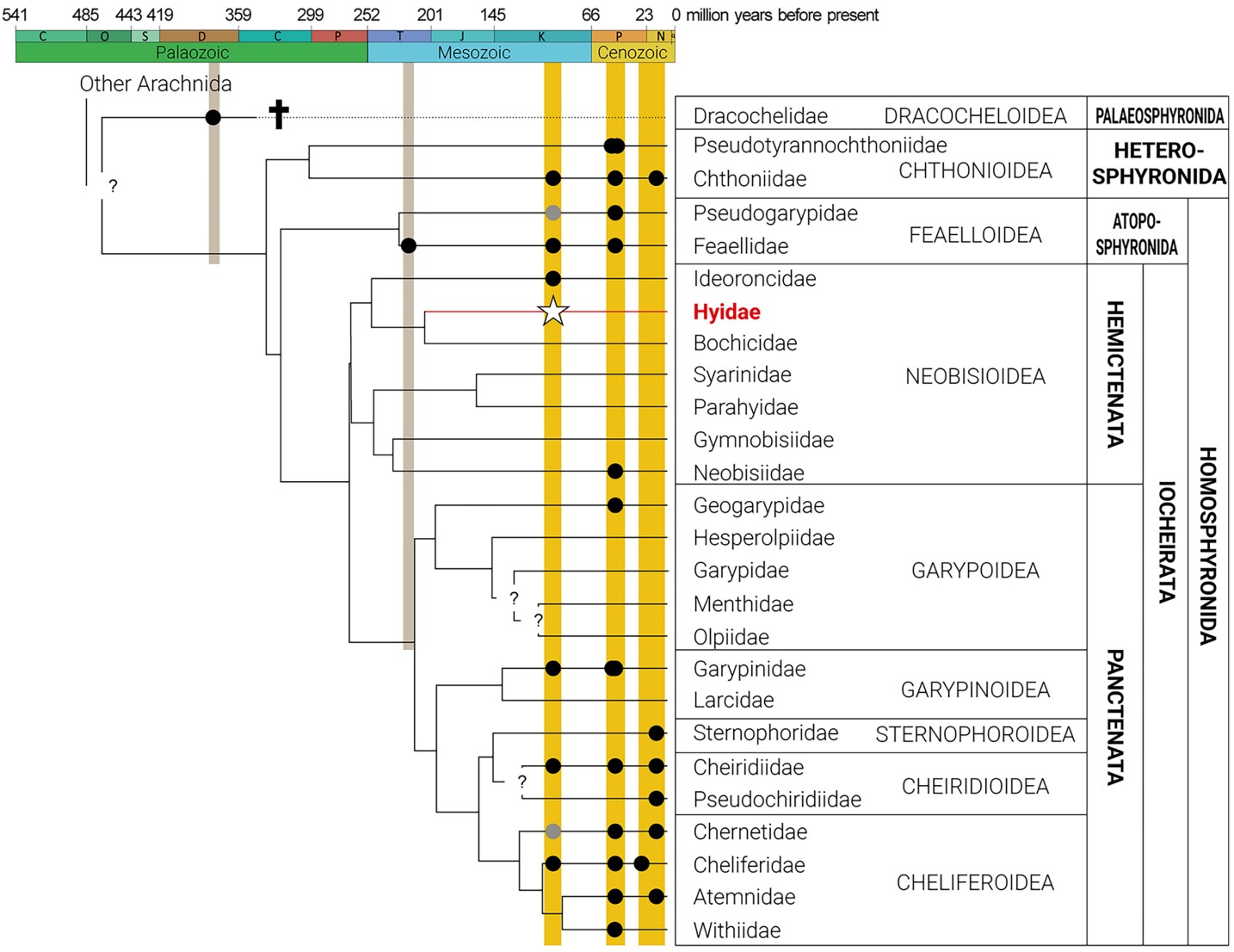

**Figure 2 Phylogeny of Pseudoscorpiones De Geer, 1778, following** *Benavides et al. (2019)***, indicating fossil taxa.** Placement of undescribed fossils (gray circles), previously described fossils (black circles), and the new fossil Hyidae *Chamberlin, 1930* species, *Hya fynni* sp. nov. (white star) indicated. Figure adapted from *Schwarze et al. (2022)*, *Geißler et al. (2022)*, *Johnson et al. (2023)*, *Novák et al. (2024)* and *Stanczak et al. (2023)*. Gray circles represent an undetermined species of Pseudogarypidae Chamberlin, 1923 from the mid-Cretaceous Rhenish Massif, Germany (*Judson, 2017*); and an undescribed species of Chernetidae Menge, 1855 from Cretaceous Canadian amber (*Schawaller, Shear & Bonamo, 1991*).

*Henderickx & Boone, 2016*; *Harms & Dunlop, 2017*; *Harvey et al., 2018*; *Porta et al., 2020*; *Wriedt et al., 2021*; *Geißler et al., 2022*; *Kolesnikov et al., 2022*; *Schwarze et al., 2022*; *Johnson et al., 2023*; *Stanczak et al., 2023*; *Novák et al., 2024*; *World Pseudoscorpion Catalog (WPC), 2024*). Among Mesozoic ambers, twelve species have been described including *Heurtaultia rossiorum Judson, 2009* in Cheliferidae Risso, 1827 from Lower Cretaceous (~102 Ma) Archingeay-Les Nouillers amber in France; *Ajkagarypinus stephani Novák et al., 2024* from Upper Cretaceous, Santonian (~86–83 Ma) Ajkaite amber in Hungary; and 10 species in one extant genus and seven extinct genera in five extant families from Burmese amber: *Electrobisium acutum Cockerell, 1917* and *Procheiridium judsoni*

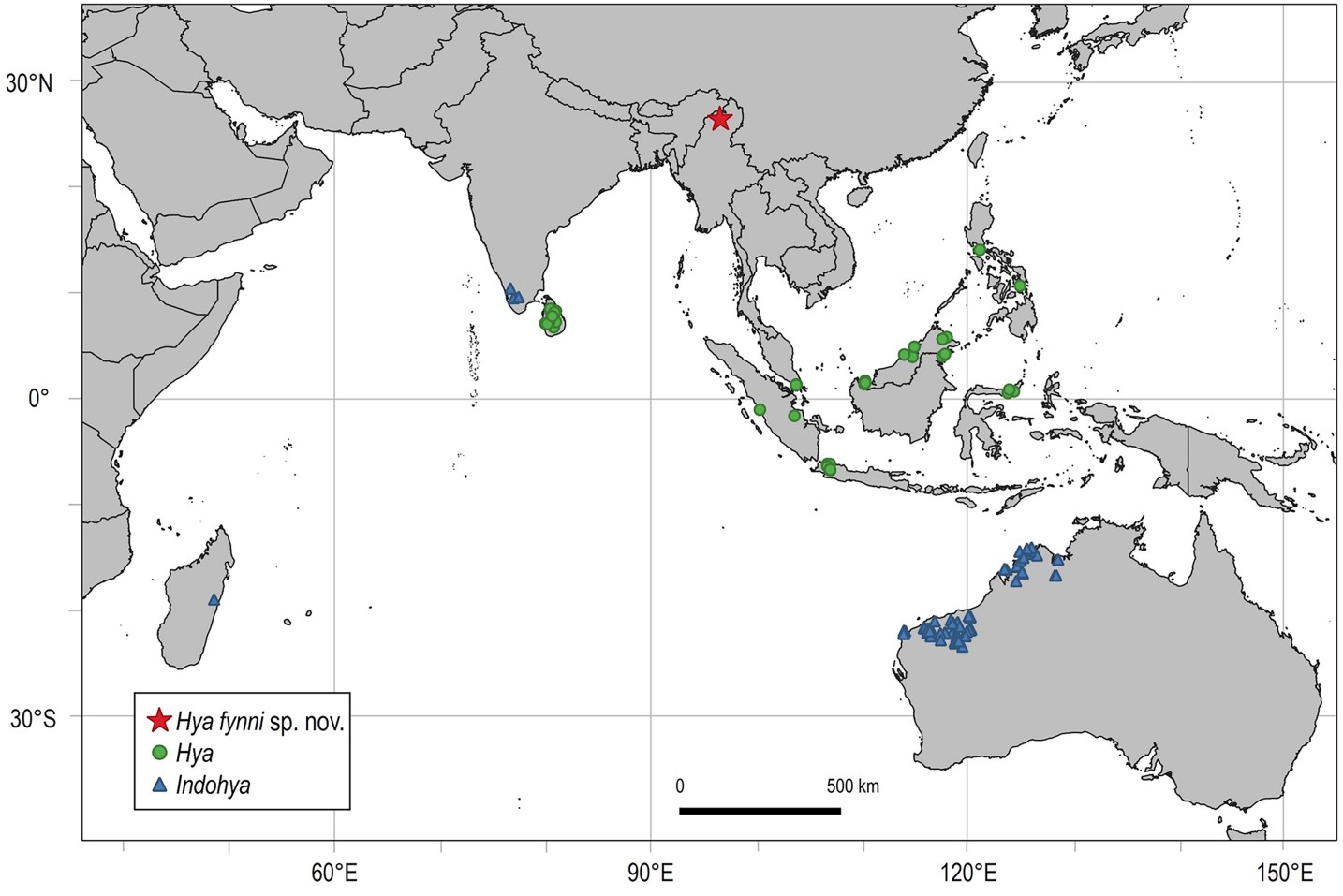

**Figure 3 Distribution of Hyidae** *Chamberlin, 1930*. Created using QGIS (v.3.16; http://www.qgis.org), by superimposing Hyidae occurrence data based on literature available from the World Pseudoscorpion Catalog (*World Pseudoscorpion Catalog (WPC), 2024*) onto political borders from Natural Earth (https://www.naturalearthdata.com).

*Porta et al., 2020* in Cheiridiidae Hansen, 1894; *Amblyolpium burmiticum* (*Cockerell, 1920*) in Garypinidae Daday, 1889; *Protofeaella peetersae Henderickx & Boone, 2016* in Feaellidae; *Weygoldtiella plausus Harvey et al., 2018*, *Prionochthonius burmiticus Wriedt et al., 2021*, *Burmeochthonius kachinae Johnson et al., 2023* and *Burmeochthonius muelleri Johnson et al., 2023* in Chthoniidae Daday, 1889; and *Proalbiorix gracilis Geißler et al., 2022* and *Proalbiorix compactus Geißler et al., 2022* in Ideoroncidae *Chamberlin, 1930* (*Cockerell, 1917*, *1920*; *Judson, 2009*; *Henderickx & Boone, 2016*; *Harvey et al., 2018*; *Porta et al., 2020*; *Wriedt et al., 2021*; *Geißler et al., 2022*; *Johnson et al., 2023*; *Stanczak et al., 2023*; *Novák et al., 2024*; *World Pseudoscorpion Catalog (WPC), 2024*). All remaining described fossil amber pseudoscorpions are only known from Cenozoic deposits (*Johnson et al., 2023*).

The pseudoscorpion family Hyidae *Chamberlin, 1930* includes 41 species in two genera, *Hya Chamberlin, 1930* and *Indohya Beier, 1974*, with a Gondwanan distribution (Fig. 3) in southwestern India, Sri Lanka, Madagascar, Southeast Asia (Indonesia, Philippines,

Singapore and Malaysia) and northwestern Australia (*Harvey, 1993*; *Harvey & Volschenk, 2007*; *Harvey et al., 2023*; *World Pseudoscorpion Catalog (WPC), 2024*). Species can be characterized as humicolous or hypogean, either living in or beneath leaf litter, under stones, or in caves in tropical forests or semi-arid deserts across their distribution (*Harvey & Volschenk, 2007*). Hyidae is unquestionably monophyletic (*Harvey & Volschenk, 2007*), and phylogenetic studies, using both morphological and molecular data, place Hyidae in the superfamily Neobisioidea in Iocheirata–a suborder characterized by the presence of venom glands in one or both chelal fingers, and the presence of a posterior maxillary lyrifissure (*Harvey, 1992*; *Murienne, Harvey & Giribet, 2008*; *Benavides et al., 2019*). Hyidae can be separated from the other six Neobisioidea families (including Bochicidae *Chamberlin, 1930*, Ideoroncidae, Syarinidae *Chamberlin, 1930*, Parahyidae *Harvey, 1992*, Gymnobisiidae *Beier, 1974* and Neobisiidae *Chamberlin, 1930*) by: the presence of a basi-dorsal mound on the femora of legs I and II that is surmounted by a small seta and slit sensilla; the extremely small setae on the anterior portion of the female genital operculum; and the presence of 2–5 stout setae on the posterior-basal margin of the pedipalpal femur (*Harvey & Volschenk, 2007*). A recent transcriptomic analysis reported an undescribed fossil *Hya* from Burmese amber that was used to help calibrate a dated evolutionary tree of Pseudoscorpiones (*Benavides et al., 2019*). According to the results, Hyidae diverged from its sister family Bochicidae in the Triassic (~210 Ma), however, the fossil piece included in this analysis was not described until now (*Ross, 2019*, *2023*).

The present contribution confirms the identification of this Cretaceous Burmese amber fossil and describes it in the genus *Hya*, thereby making it the first described fossil species for Hyidae and the third described Burmese fossil for Neobisioidea. The new fossil species joins *A. burmiticum* in being the oldest records for extant pseudoscorpion genera. Additionally, this contribution proposes a biogeographical hypothesis for Hyidae's current distribution.

## MATERIALS AND METHODS

The amber piece studied here is deposited in the Key Lab of Insect Evolution and Environmental Changes, College of Life Sciences, Capital Normal University, Beijing, China (CNUB; Dong Ren, Curator). The piece was purchased by Mr. Fangyuan Xia in 2015 and donated for this study in the same year.

In preparation for imaging, the amber piece was wet-sanded with carbide sandpaper of different grain sizes (Waterproof *SiC*, FEPA P #120, #1200, #4000, Struers GmbH) and coated with a single-component polyurethane resin (Acrüdur R40; Adolf C.C. Rüegg GmbH & Co. KG, Rapperswil-Jona, Switzerland) for protection following *Frenzel, Kotthoff & De Francesco Magnussen (2021)* and *Hoffeins (2022)*. Photographs were taken using a microptic stacking BK PLUS Lab System (Dun, Inc.) mounted with a Canon EOS 7D Mark II Camera at 5×, 10× and 20× magnification. Images were processed with Capture One Pro 9.3 (v.9.3.0.85) and stacked with Zerene Stacker (v.1.04; Zerene Systems LLC, Richland, WA, USA). The amber piece was submerged in baby oil (Bübchen Baby Öl, Bübchen Werk Ewald Hermes, Pharmazeutische Fabrik GmbH) and surrounded by a

diffusor to optimize imaging. Images were edited in Adobe Photoshop (v.25.1.0). Standard pseudoscorpion measurements (in mm) following *Chamberlin (1930)* and *Romero-Ortiz, Flórez & Sarmiento (2020)* were taken using a Leica M205 A stereomicroscope armed with a 0.63x (Leica 10450027) lens, Leica DMC4500 camera and Leica Application Suite X (v.3.7.5.24914) software.

Synchrotron radiated micro-computed tomography (SRµCT) was performed at the imaging beamline P05 (IBL) operated by Helmholtz-Zentrum Hereon at PETRA III at Deutsches Elektronen-Synchrotron in Hamburg, Germany (*Greving et al., 2014*; *Wilde et al., 2016*). Scanning occurred using a custom-built CMOS camera (*Lytaev et al., 2014*) with a photon energy of 18 keV. Raw projections were binned twice, and a reconstruction was made by applying a transport of intensity phase retrieval approach using the Filtered Back Project algorithm, implemented in a custom reconstruction pipeline (*Moosmann et al., 2014*) with Matlab (Math-Works) and Astra Toolbox (*Palenstijn, Batenburg & Sijbers, 2011*; *Van Aarle et al., 2015*, *2016*). Segmentation was performed in Amira (v.6.0.1; FEI Company, Hillsboro, OR, USA) by selecting the specimen in every 20th or 30th image to generate a label, which was then used to interpolate all intervening images with the program Biomedisa (*Lösel et al., 2023*: https://biomedisa.org/). A 3D-rendering was then exported and deposited in MorphoSource (https://www.morphosource.org/; Project ID: 000583128).

Locality records of the family Hyidae were compiled from the World Pseudoscorpion Catalog (*World Pseudoscorpion Catalog (WPC), 2024*) and georeferenced if coordinates were not provided. A distribution map, including all published records for the family, was created using QGIS (v.3.16; http://www.qgis.org), by superimposing occurrence data onto political borders from Natural Earth (https://www.naturalearthdata.com). All figures were prepared and edited using Adobe Illustrator (v.28.0), Corel Draw X8 or Concepts (Pro Version 2024.5.3).

The electronic version of this article in Portable Document Format (PDF) will represent a published work according to the International Commission on Zoological Nomenclature (ICZN), and hence the new names contained in the electronic version are effectively published under that Code from the electronic edition alone. This published work and the nomenclatural acts it contains have been registered in ZooBank, the online registration system for the ICZN. The ZooBank LSIDs (Life Science Identifiers) can be resolved and the associated information viewed through any standard web browser by appending the LSID to the prefix http://zoobank.org/. The LSID for this publication is: urn:lsid:zoobank.org: pub:9BD696DF-FD28-4D0A-ABA5-F2DFAD49A28E. The online version of this work is archived and available from the following digital repositories: PeerJ, PubMed Central SCIE and CLOCKSS.

### Systematics

Order Pseudoscorpiones de Geer, 1778
Suborder Iocheirata *Harvey, 1992*
Infraorder Hemictenata Balzan, 1892
Superfamily Neobisioidea *Chamberlin, 1930*

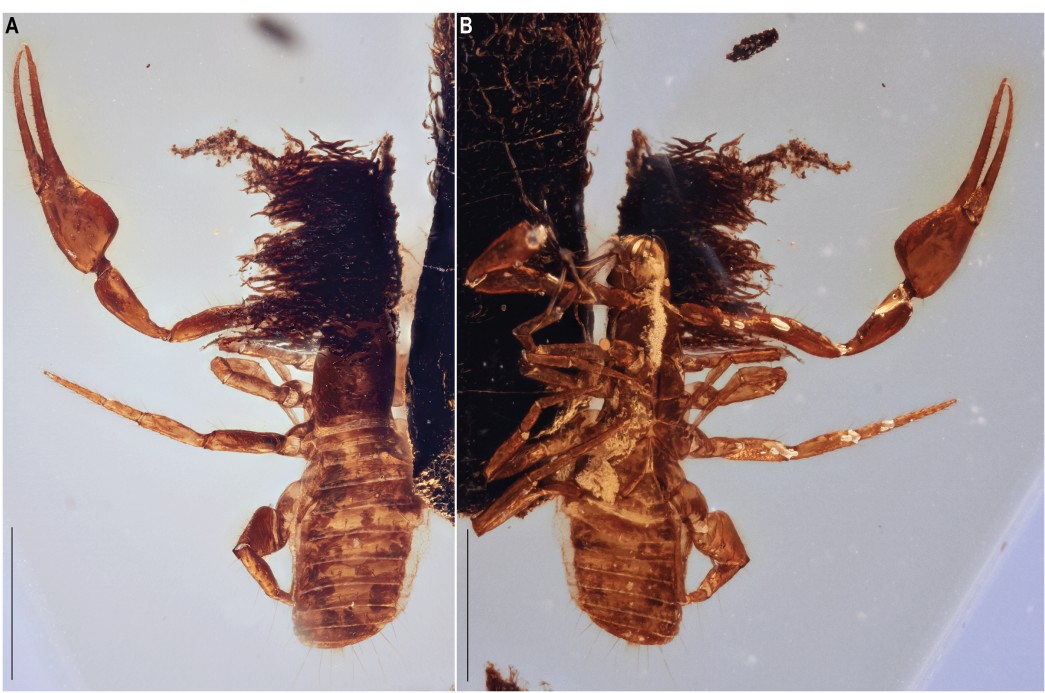

**Figure 4** *Hya fynni* sp. nov., holotype ♀ (CNU-PSE-MA2016010), dorsal (A) and ventral (B) habitus. Scale bars = 0.5 mm.

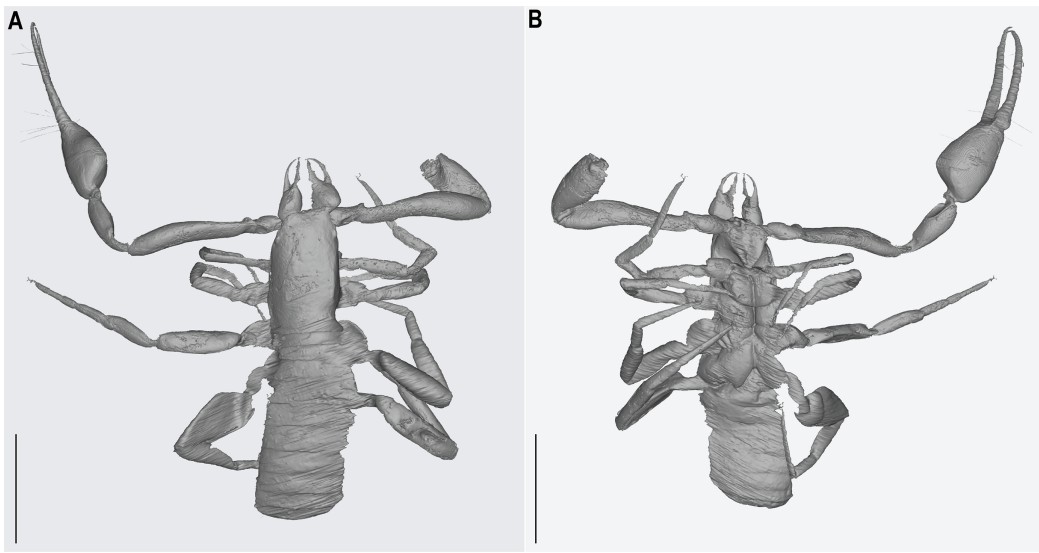

**Figure 5** *Hya fynni* sp. nov., holotype ♀ (CNU-PSE-MA2016010), 3D-reconstruction model, dorsal (A) and ventral (B) habitus. Scale bars = 0.5 mm.

Family Hyidae *Chamberlin, 1930*
Genus *Hya Chamberlin, 1930*

*Hya fynni* **sp. nov.**
Figures 2–7, 8A

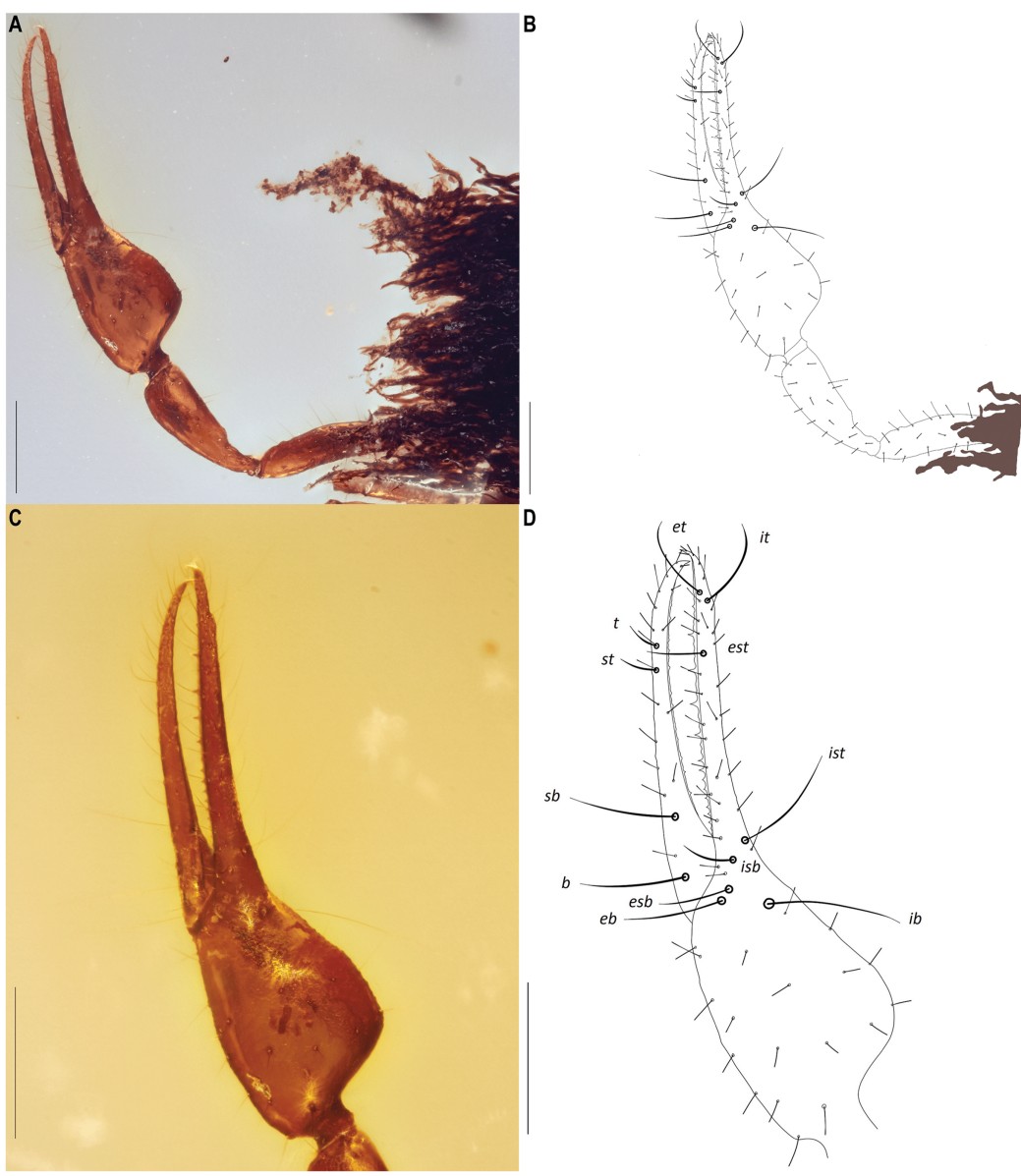

**Figure 6 *Hya fynni* sp. nov., holotype ♀ (CNU-PSE-MA2016010), dorsal left pedipalp (A, B) and dorsal left chela (C, D) as photographs (A, C) and drawings (B, D).** Scale bars: (A, B), 0.5 mm; (C, D), 0.2 mm.

ZooBank: urn:lsid:zoobank.org:act:01185A07-5675-40AB-A3DC-27E66924D8DF.
MorphoSource: doi: 10.17602/M2/M589333
*Hya* sp.: *Benavides et al. (2019)*: Table 3; *Harvey et al. (2023)*: 5.
Type Material: Holotype, adult female (CNUB-PSE-MA2016010), Burmese amber.
Locality & Horizon: Hukawng Valley (26°20′N 96°36′E), Kachin, Myanmar; lowermost Cenomanian (ca. 99 Ma), Late Cretaceous.
Preservation: The specimen is preserved in a nearly translucent amber piece. Although the specimen is well preserved, some features are not visible dorsally including, the right legs, base of the left pedipalp and anterior portion of the carapace (eyes and chelicerae). Only

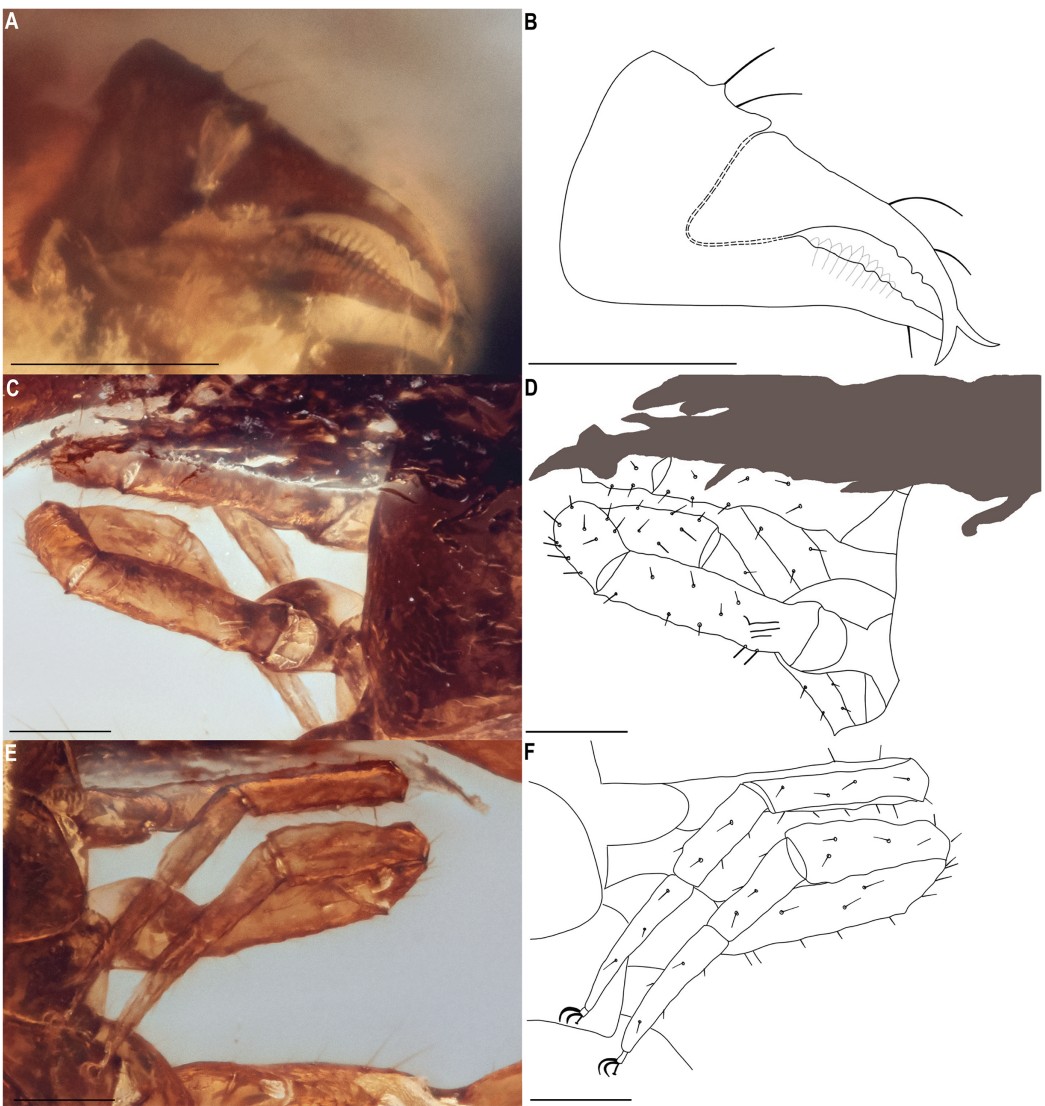

**Figure 7** *Hya fynni* **sp. nov., holotype ♀ (CNU-PSE-MA2016010), chelicerae (A, B), and left legs I and II (C–F), as photographs (A, C, E) and drawings (B, D, F).** (A, B) Chelicerae. (C, D) Legs I and II, dorsal. (E, F) Legs I and II, ventral. Scale bars = 0.1 mm.

the left chela is present which, due to the cut of the amber, cannot be seen directly dorsally, ventrally, or laterally, but only from the dorsolateral or ventrolateral views. Chelicerae are visible in ventral view (Fig. 4B), and the chelicerae, eyes and obscured pedipalpal regions are revealed by the synchrotron scan (Fig. 5).

Etymology: This specific epithet, 'fynni' is a patronym honoring the first author's son, Fynn Röschmann, who was born during the time the description was prepared.

Diagnosis: *Hya fynni* sp. nov. can be separated from *Hya minuta* (*Tullgren, 1905*) and *Hya chamberlini Harvey, 1993* by the more proximal position of trichobothrium *est* on the fixed finger (Figs. 6, 8). In *H. minuta* and *H. chamberlini*, trichobothrium *est* is situated more distally on the fixed finger, and is only ~40 μm proximal from trichobothrium *it* (Fig. 8B). This contrasts with *H. fynni* sp. nov. in which trichobothrium *est* is situated more

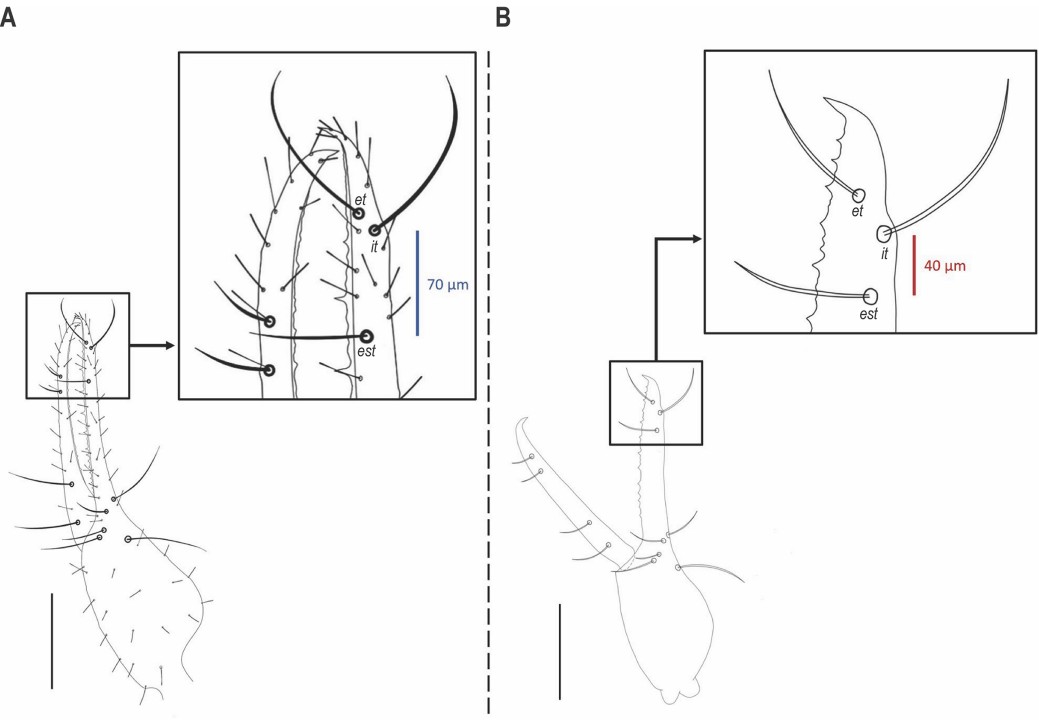

**Figure 8 Position of trichobothrium *est* on pedipalpal chela in *Hya* species.** (A) *Hya fynni* sp. nov., holotype ♀ (CNU-PSE-MA2016010). (B) *Hya minuta* (*Tullgren, 1905*). Scale bars = 0.2 mm.

proximally on the fixed finger, and is ~70 μm proximal from trichobothrium *it* (Fig. 8A). *H. fynni* is also larger (chela length with pedicel, ♀: 0.78) than *H. minuta* (chela length with pedicel, ♀: 0.68–0.86) and *H. chamberlini* (chela length with pedicel, ♀: 0.56–0.67). Description: The following description is based on the holotype, the only known specimen. The specimen is definitively adult given that the complete number of trichobothria are present and likely female.

*Color* (in amber): Uniform brown.

*Chelicerae* (Fig. 7A): Only visible in ventral view. Two setae on retrolateral surface of chelal hand and two on retrolateral surface of movable finger; galea unbranched, long and acute; teeth barely visible; rallum not visible.

*Pedipalp*s (Figs. 6, 8A): Long and stout; chelal hand 1.45× longer than broad, chelal finger 1.52× longer than hand. Chelal hand and fixed finger with eight trichobothria: *et*, *it* and *est* situated distally on fixed finger; *ist*, *isb*, *esb* and *eb* positioned proximally on fixed finger; *ib* located on dorsal surface of chelal hand. Four trichobothria on movable finger: *t* and *st* distally, *sb* and *b* proximally. Fixed finger with 14 chelal teeth, heterodentate, closely spaced proximally, but widely spaced distally; chelal teeth on movable finger present. Stout setae not visible on pedipalpal femur. Venom apparatus not clearly identifiable but venom teeth visible in both fingers.

*Prosoma* (Fig. 4): Carapace rectangular, 2.43× longer than broad. Setae, eyes and genitalia not visible.

*Opisthosoma* (Fig. 4): Tergites and sternites uniform without medial suture. Tergal chaetotaxy: 4: 4: 4: 4: 6: 7: 7: 7: 8: 6 (not clearly visible).

*Legs* (Figs. 7C–7F): Femora I and II with basi-dorsal mound and two small setae; three slit sensilla visible on left femur II. Arolium shorter than claws.

*Dimensions*: Body length 1.25; chelicera 0.19/0.1; carapace 0.56/0.23; opisthosoma 0.69/0.42. Pedipalps: trochanter 0.14/0.08, femur 0.39/0.08, patella 0.29/0.1, chela length (without pedicel) 0.75, chela length (with pedicel) 0.78/0.21, hand length 0.32, movable finger length 0.45. Left leg I: trochanter 0.06/0.07, femur 0.14/0.04, patella 0.08/0.05, tibia 0.15/0.03, metatarsus 0.11/0.03, tarsus 0.17/0.02; left leg II: trochanter 0.12/0.09, femur 0.16/0.06, patella 0.08/0.05, tibia 0.15/0.05, metatarsus 0.1/0.04, tarsus 0.19/0.03; left leg III: trochanter 0.1/0.1, femur-patella 0.26/0.06–0.07, tibia 0.2/0.05, metatarsus 0.13/.0.4, tarsus 0.23/0.03; left leg IV: trochanter 0.09/0.06, femur-patella 0.33/0.09–0.15, tibia 0.16/0.07, metatarsus 0.1/0.03, tarsus 0.26/0.03.

Other Inclusions: The specimen is surrounded by plant residues, which obscure parts of the body.

## Key to species in the genus *Hya*

1. Trichobothrium *est* on fixed finger situated ~70 μm proximal to *it*…….*Hya fynni* sp. nov.

Trichobothrium *est* on fixed finger situated ~40 μm proximal to *it*……………………..2

2. Chela (with pedicel) 0.605–0.80 (♂), 0.68–0.85 (♀) mm in length; 4–5 setae anterior to *ib*……………………………………………………………*Hya minuta* (*Tullgren, 1905*)

Chela (with pedicel) 0.48–0.615 (♂), 0.56–0.67 mm (♀) in length; 1 seta anterior to *ib* (often absent)……………………………………….…*Hya chamberlini* *Harvey, 1993*

# DISCUSSION

## Systematic position of *H. fynni*

The pseudoscorpion family Hyidae was first erected by *Chamberlin (1930)* based on a single species, *Hya heterodentata Chamberlin, 1930* from Luzon Island in the Philippines. *Tullgren (1905)* originally described *Ideobisium minutum Tullgren, 1905* from Java, Indonesia in the family Obisiidae Sundevall, 1833 (=Neobisiidae), but this species was transferred to *Hya* by *Beier (1974)*, who described a second hyid genus, *Indohya* from southwestern (Kerala) India. *Harvey (1993)* synonymized *H. heterodentata* with *H. minuta*, thereby making *H. minuta* the type species of Hyidae; described *H. chamberlini* from Sri Lanka; placed *Hya* in its own subfamily Hyinae *Harvey, 1993*; and described five new species in *Indohya* and *Hyella Harvey, 1993* from northwestern Australia, placing both genera in the subfamily Indohyinae *Harvey, 1993*. The first phylogenetic analysis of the family, based on 57 morphological characters scored for 14 hyid species, consistently recovered Hyidae and *Hya* as monophyletic (*Harvey & Volschenk, 2007*), but found *Indohya* to be rendered paraphyletic by *Hya*, and *Hyella* nested within *Indohya*. Based on this analysis, *Hyella* was designated a junior synonym of *Indohya*, and five new *Indohya*

species were described, including the first family record for Madagascar. *Harvey et al. (2023)* performed a phylogenetic analysis for all *Indohya* species from Australia using both morphological and molecular (COI, 28S rDNA and 18S rDNA) data, and added an additional 27 new species, bringing the total number of extant species in the family to 41. The molecular analysis also found that two populations (Brunei and Singapore) identified as *H. minuta* had large barcode differences, suggesting the presence of cryptic species (*Harvey et al., 2023*).

The new fossil species can confidently be assigned to the family Hyidae based on the presence of trichobothrium *ib* on the dorsal margin of the left chelal hand (Fig. 6), and the presence of slit sensilla and two setae on the basi-dorsal mound of the femora of left leg II (Figs 7C, 7D). On the posterior surface of the left pedipalpal femur, a few stout setae are somewhat visible, however, it is impossible to determine with confidence if these represent the same setae diagnostic for the family. The genitalia are not visible and therefore sexing *H. fynni* is difficult, however, the specimen is large (1.25 mm) and is likely an adult female. In ascribing the fossil to the genus *Hya*, attention is paid to the chelal teeth, venom apparatus and small setae anterior to trichobothrium *ib*. As is characteristic of *Hya*, the venom apparatus is present in both chelal fingers; the chelal teeth are heterodentate and widely spaced; and two small setae are positioned anteriorly to trichobothrium *ib*. This contrasts with species of *Indohya*, in which the venom apparatus is present only in the movable finger; the chelal teeth are homodentate and closely spaced; and setae anterior to *ib* are absent. Although morphologically similar to extant *Hya* species, *H. fynni* can be readily separated from *H. minuta* and *H. chamberlini*, based on the position of trichobothrium *est*, which is more proximal in *H. fynni* (70 µm from *it*), than in *H. minuta* and *H. chamberlini* (40 µm from *it*) (Fig. 8). *H. fynni* is also the largest *Hya* species (total body length, ♀: 1.25; chela length with pedicel, ♀: 0.78), followed by *H. minuta* (chela length with pedicel, ♀: 0.68–0.86), and *H. chamberlini* (chela length with pedicel, ♀: 0.56–0.67).

*H. fynni* is the first described fossil species of the family Hyidae and the third described Burmese fossil of the superfamily Neobisioidea (Fig. 2). Moreover, *H. fynni* represents the second Cretaceous fossil record for an extant pseudoscorpion genus. The oldest fossil record for an extant pseudoscorpion family is *A. henderickxi* in the family Feaellidae, which dates to the Upper Triassic (Upper Carinan–Lower Norian: ~227 Ma) (*Kolesnikov et al., 2022*). *Hya fynni* now joins *Amblyolpium burmiticum* (also from Burmese amber) (*Cockerell, 1920*; *Judson, 2009*) in being the oldest fossil records for extant pseudoscorpion genera. In their dated molecular phylogeny of Pseudoscorpiones, *Benavides et al. (2019)* proposed that with the exception of Chthonioidea, which originated in the Paleozoic, all other Pseudoscorpion families were established in the Mesozoic. Within Iocheirata (pseudoscorpions with venom glands), the Neobisioidea families, including Hyidae, were inferred to have originated in the Triassic–Jurassic, whereas those in Panctenata are presumed to have originated somewhat later in the Jurassic–Cretaceous (*Benavides et al., 2019*). Until now, only eight families were confirmed through fossils to exist by the Cretaceous: Cheiridiidae, Chthoniidae, Feaellidae, Garypinidae and Ideoroncidae from Burmese amber; Cheliferidae from Archingeay amber; and one undescribed species each of

Pseudogarypidae Chamberlin, 1923 and Chernetidae Menge, 1855 from the mid-Cretaceous Rhenish Massif and Canadian amber, respectively (*Judson, 2009*; *Ross, 2019*, *2023*; *Geißler et al., 2022*). *Hya fynni* sp. nov. confirms that both Hyidae and its extant genus *Hya* existed by the Cretaceous (~99 Ma). With the exception of its slightly larger size and the position of trichobothrium *est* on the pedipalpal chela, this extinct species appears morphologically identical to its extant relatives, *H. minuta* and *H. chamberlini*—an extreme case of morphological stasis over a 99-million-year period. These results also echo previous studies on fossil pseudoscorpions, which have found similar cases of morphological stasis (*Geißler et al., 2022*; *Johnson et al., 2023*; *Stanczak et al., 2023*).

## Late rifting of the Burma Terrane?

The BT rifted from Gondwana, drifted northwards and collided with Eurasia (Fig. 1A) (*Heine, Müller & Gaina, 2004*; *Seton et al., 2012*; *Metcalfe, 2011*, *2013*; *Zhang et al., 2017*; *Westerweel et al., 2019*). The timing of these events, and the position of the BT in relation to other landblocks during its journey, however, is still debated. Various hypotheses have been proposed based on different types of data, including palaeomagnetic, and plant and animal fossils. An "*Early Devonian Rifting*" hypothesis suggests that the BT rifted from Gondwana in the Early Devonian, and travelled with other continental blocks (South China, West Sumatra, Indochina and East Malaya) as a microcontinent known as "Cathaysialand" in the Permian, opening up the Palaeo-Tethys, until finally reaching Eurasia in the Late Triassic/Early Jurassic (*Metcalfe, 2011*, *2013*). This hypothesis is supported by the fossil fauna, which is said to resemble that of Gondwana in the Devonian, but which already showed differences in the Carboniferous (*Metcalfe, 2011*, *2013*). Alternatively, the "*Late Jurassic Rifting*" hypothesis suggests that the BT separated from Gondwana in the Late Jurassic and collided with Sundaland either in the Late Cretaceous (*Heine, Müller & Gaina, 2004*; *Seton et al., 2012*; *Zhang et al., 2017*), or even as late as the Eocene or Oligocene (*Westerweel et al., 2019*). Support for a later separation comes from angiosperms (*Poinar, 2018*), wasps (*Jouault, Perrichot & Nel, 2021*), short-tailed whip-scorpions (*De Francesco Magnussen et al., 2022*), spiders (*Wood & Wunderlich, 2023*) and ticks (*Chitimia-Dobler et al., 2023*) preserved in Burmese amber, which rule out a Devonian separation. Instead, they suggest that the BT could not have separated from Gondwana before the Mesozoic (*Poinar, 2018*; *Jouault, Perrichot & Nel, 2021*; *De Francesco Magnussen et al., 2022*; *Chitimia-Dobler et al., 2023*; *Wood & Wunderlich, 2023*).

The present-day distribution of Hyidae reflects that of former Gondwanan lineages, and it was already proposed that this family originated on Gondwana and was shaped by Mesozoic vicariance (*Harvey, 1993*, *1996a*, *1996b*; *Harvey & Volschenk, 2007*). Within *Indohya*, all species are found on landmasses (Madagascar, southwestern India, western Australia) that were once adjacent to each other, and phylogenetic analyses suggest that *Indohya* was distributed across these contiguous regions in the Early Jurassic, and diversified during Gondwanan breakup by the Late Jurassic–Cretaceous (*Harvey, 1996b*; *Harvey & Volschenk, 2007*). Most *Indohya* species are also considered to be short-range endemics (*Harvey et al., 2023*), thereby making dispersal over long distances unlikely and

providing support for vicariant speciation (*Harvey & Volschenk, 2007*), but within the genus *Hya*, the vagility of species varies. *H. chamberlini* is restricted to Sri Lanka, a proposed former refugium during Deccan Trapp volcanism (*e.g.*, *Conti et al., 2002*; *Loria & Prendini, 2020*). This contrasts with *H. minuta*, the most widespread species in Hyidae, which is found across Southeast Asia in Peninsular Malaysia, and the islands of Sumatra, Java, Borneo, Sulawesi, Luzon and Leyte, but does not reach the Moluccas Islands to the east (Fig. 3). However, *Harvey et al. (2023)* reported large differences in the barcoding gene COI in populations of *H. minuta* from Brunei and Singapore, suggesting previously undetected cryptic speciation. The discovery of *H. fynni* in Burmese amber not only expands the paleo-distribution of Hyidae to another landmass (*i.e.*, the BT), but also confirms that *Hya* was present on the BT by the Cenomanian (99 Ma) and had diversified by this time. Considering the proposed Late Triassic origin (~210 Ma) for Hyidae, the "*Late Jurassic Rifting*" hypothesis seems more plausible (Fig. 1A) to explain the current distribution of *Hya*, and this hypothesis was also preferred in previous studies on Burmese amber pseudoscorpions, including the family Ideoroncidae (*Geißler et al., 2022*) and tribe Chthoniini (Chthoniinae) (*Wriedt et al., 2021*). Alternatively, the "*Early Devonian Rifting*" hypothesis was favored to explain the presence of Burmese ambers fossils in the tribe Tyrannochthoniini (Chthoniinae) based on previously published divergence dates (*Benavides et al., 2019*; *Johnson et al., 2022*, *2023*). However, in the absence of biogeographical analyses, these remain speculations, and future research should focus on the systematic placement of Burmese fossil pseudoscorpions within dated phylogenies.

## Ecological stasis

The Cretaceous was one of the hottest periods, and fossil evidence generally supports a warm climate and tropical/sub-tropical forest with some freshwater habitats on the BT during this time (*Grimaldi, Engel & Nascimbene, 2002*; *Poinar, 2018*; *Xing et al., 2018*). Among arachnids preserved in Burmese amber, all of their modern counterparts include species that can be found today in tropical rainforests (*Selden & Ren, 2017*). Although amber is generally biased towards preserving corticolous (bark-dwelling) arthropods, many probable humicolous (litter-dwelling) species have also been documented (*e.g.*, *Dunlop & de Oliveira Bernardi, 2014*; *Müller et al., 2020*; *De Francesco Magnussen et al., 2022*). Based on the ecology of extant relatives, seven Burmese amber pseudoscorpion species including, the chthoniids *B. muelleri*, *B. kachinae*, *P. burmiticus* and *W. plausus*, the feaellid *P. peetersae*, and the ideoroncids *P. gracilis* and *P. compactus*, were likely humicolous, whereas the cheiridiid *P. judsoni* was possibly corticolous like many of its Recent relatives (*Geißler et al., 2022*; *Johnson et al., 2023*). The ecology of the garypinids, *E. acutum* and *A. burmiticum*, is less certain given that extant relatives are found in more variable microhabitats that include both tree bark and leaf litter.

Extant species of Hyidae are found in equatorial or low-latitudinal (<30°N/S) regions. *Hya* is exclusively found in tropical rainforests, whereas *Indoyhya* inhabits both monsoon rainforest and gullies/caves in the semi-arid Pilbara region of Western Australia. Within *Indohya*, species are classified as humicolous or hypogean (cave-dwelling), with most hypogean species exhibiting troglomorphic features, such as elongated appendages and eye

loss (*Harvey et al., 2023*). *Hya minuta* and *H. chamberlini* are both humicolous species, and have been recorded in the soil or under leaf litter across their distribution (*Harvey & Volschenk, 2007*). *Hya fynni* is nearly indistinguishable from these modern species, and lacks troglomorphic characters. Given all available evidence, the most parsimonious situation would assume that *H. fynni*, like all extant *Hya* species, was living in humicolous environments in a tropical rainforest on the BT. This also implies that the ecology of *Hya* has not changed in the last 99 million years, and marks an exceptional case not only of morphological stasis but also ecological niche stasis of an arthropod lineage over an extensive period. This also contrasts with data available for some other arachnid lineages, such as the araneoid spider fauna (*Wunderlich, 2008*; *Magalhaes et al., 2020*) that has seen major changes since the Cretaceous, and even some other niche-conserved arachnids such as Ricinulei (*Botero-Trujillo et al., 2022*).

## CONCLUSIONS

The present contribution describes the first fossil species of Hyidae and the third Burmese fossil species in the superfamily Neobisioidea. Based on morphology, the new fossil, *H. fynni* is readily placed in the genus *Hya*, providing evidence that the family Hyidae and the genus *Hya* already existed by the Cretaceous—making it, and *Amblyolpium burmiticum*, the oldest fossil records for extant pseudoscorpion genera. *Hya fynni* closely resembles the Recent species, *H. minuta* and *H. chamberlini*, and only differs in the position of trichobothrium *est* on the pedipalpal chela, supporting long-term morphological stasis within the genus. Moreover, considering the proposed divergence dates for Hyidae, the newly described species supports a Gondwanan origin for Hyidae, and the "*Late Jurassic Rifting*" hypothesis, in which the BT rifted from Gondwana in the Late Jurassic and collided by the Cretaceous/Eocene. Given support for a tropical rainforest on the BT during the Cretaceous and the occurrence of *H. minuta* and *H. chamberlini* in humicolous environments of tropical rainforest today, it can be assumed that *H. fynni* inhabited similar habitats as extant species, implying that the ecology of *Hya* has remained unchanged for the last 99 million years.

## ACKNOWLEDGEMENTS

We are grateful to Jithin Johnson for assistance with morphological characters. We also thank colleagues at PETRAIII at the German Electron Synchrotron DESY (Deutsches Elektronen-Synchrotron) and student helpers for assistance with scanning, as well as Jason Dunlop and an anonymous reviewer for helpful feedback on the manuscript.

### Funding

Funding was provided by the German Science Foundation award HA 8785/5 and KO 3944/10 to Danilo Harms and Ulrich Kotthoff. Synchrotron scans were generated with support of a DESY Block Allocation Group proposal (BAG-20190010) "Scanning the past–Reconstructing the diversity in million year old fossil amber specimens using SRμCT"

at PETRA III at DESY, a member of the Helmholtz Association. This research was also supported through the Maxwell computational resources operated at Deutsches Elektronen-Synchrotron DESY, Hamburg, Germany. There was no additional external funding received for this study. The funders had no role in study design, data collection and analysis, decision to publish, or preparation of the manuscript.

### Grant Disclosures

The following grant information was disclosed by the authors:
German Science Foundation award: HA 8785/5 and KO 3944/10.
DESY Block Allocation Group proposal: BAG-20190010.
Deutsches Elektronen-Synchrotron DESY, Hamburg, Germany.

### Competing Interests

The authors declare that they have no competing interests.

### Author Contributions

- Liza M. Röschmann performed the experiments, analyzed the data, prepared figures and/or tables, authored or reviewed drafts of the article, and approved the final draft.
- Mark S. Harvey conceived and designed the experiments, performed the experiments, analyzed the data, authored or reviewed drafts of the article, and approved the final draft.
- Yanmeng Hou analyzed the data, authored or reviewed drafts of the article, and approved the final draft.
- Danilo Harms conceived and designed the experiments, authored or reviewed drafts of the article, and approved the final draft.
- Ulrich Kotthoff conceived and designed the experiments, authored or reviewed drafts of the article, prepared figures and/or tables and approved the final draft.
- Jörg U. Hammel performed the experiments, authored or reviewed drafts of the article, and approved the final draft.
- Dong Ren conceived and designed the experiments, authored or reviewed drafts of the article, and approved the final draft.
- Stephanie F. Loria conceived and designed the experiments, performed the experiments, analyzed the data, prepared figures and/or tables, authored or reviewed drafts of the article, and approved the final draft.

### Data Availability

The 3D-rendering is available in MorphoSource: DOI 10.17602/M2/M589333.

### New Species Registration

The following information was supplied regarding the registration of a newly described species:
Publication LSID: urn:lsid:zoobank.org:pub:9BD696DF-FD28-4D0A-ABA5-F2DFAD49A28E.

Hya fynni species LSID: urn:lsid:zoobank.org:act:01185A07-5675-40AB-A3DC-27E66924D8DF.

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
