# Peer review of "First fossil species of family Hyidae (Arachnida: Pseudoscorpiones) confirms 99 million years of ecological stasis in a Gondwanan lineage"

_PeerJ, doi:10.7717/peerj.17515_

## Round 0.1 · original submission · Minor Revisions

Please, address all minor comments of the reviewer either directly in the text or as an answer in a separate file.

·

Basic reporting

This is a nice paper describing a new species of pseudoscorpion arachnid from Burmese amber. The author team includes acknowledged experts on both modern and fossil pseudoscorpions, as well as experts on the Burmese amber and its fauna. The manuscript is very well written throughout, is structured correctly for a standard systematic description of a new species and the figures (including some good synochrotron images) are of high quality. The references are extensive and appear up to date. Overall, I would recommend publication subject to MINOR REVSION.

Jason A. Dunlop, Berlin

Experimental design

I believe this manuscript falls within the scope of the journal, and represents a valuable contribution to arachnid palaeontology and our wider understanding of the Burmese amber fauna and its biogeographical origins.This deposit now represents probably the most important source of data for terrestrial arthropods in the Cretaceous. The particular significance of the present study is the fact that the authors report a new pseudoscorpion family for the Cretaceous, representing both the oldest and only fossil record of this particlar family, and can even place it within a living SE Asian genus. This implies a considerable degree of both morphological, and indeed ecological, stasis for this particular genus and family.

There is no experiment per se, but the systematic description has been carried out to a very high standard by experienced pseudoscorpion workers. The fossil in question has been deposited in a recognised public collection in Beijing and should thus be available for later restudy if necessary. Data from the description has been uploaded to Morphobase and the new species has been registered with Zoobank.

Validity of the findings

The authors interpretation of the significance of their fossil is excellent and it fits into a wider emerging pattern seen in other arachnid/arthropod groups (see also Additional comments) that the Burmese amber fauna may have largely Gondwanan origins. Figures 2 and 3 summarising respectively the geographical distribution of the family today and its position in the overall pseudoscorpion evolutionary tree are especially useful and relevant here for understanding the significance of their discovery. All necessary data for identifiying the fossil appear to be present in the descriptions and figures.

Additional comments

Minor comments:

Line 206: You might want to clarify the visibility statement here and start by saying what can be seen using standard optical methods. Chelicerae are visible in the ventral view (Fig. 4B) and chelicerae, eyes and part of the 'missing' pedipalp are revealed by the synchrotron scan (Fig. 5).

On line 303 you mention "Although at least one other undescribed fossil Hyidae is known (Xia et al., 2015; Harms & Dunlop, 2017;)" is this the same specimen Benevides et al. were referring to in their 2019 paper - i.e. the one you describe here? - or is this an additional specimen in which case you may want to clarify that it is also a (figured?) example in Burmese amber? Perhaps in a private collection?

Line 304: delete ";" after 2027

Line 314: you might want to be a little more circumspect here in the absence of fossil data and write something like "Neobisioidea families, including Hyidae, were inferred to have originated in the Triassic-Jurassic, whereas those in Panctenata are presumed to have originated somewhat later in the Jurassic-Cretaceous (Benavides et al., 2019)."

Line 346: In addition to the angisperm data, you might want to place your results is a wider context and check if (other) arthropods support Gonwanan origins for the Burmese amber fauna. Off the top of my head you might consider for arachnids:

Wood, H. M., & Wunderlich, J. (2023). Burma Terrane amber fauna shows connections to Gondwana and transported Gondwanan lineages to the Northern Hemisphere (Araneae: Palpimanoidea). Systematic Biology, syad047.

Chitimia-Dobler, L., Dunlop, J. A., Pfeffer, T., Würzinger, F., Handschuh, S., & Mans, B. J. (2023). Hard ticks in Burmese amber with Australasian affinities. Parasitology, 150, 157-171.

Both palpimanoid spiders, as part of the more derived araneomorphs, and ixodid ticks, as parasites of fully terrestrial vertebtates, are unlikley to have been crawling around in the Devonian, and probably fit better to a later (i.e. Jurassic/Cretaceous?) rifting event.

Line 378: better "The Cretaceous was one of the hottest periods in Earth history..." ?

Line 381: "all modern forms are found in tropical rainforest". Should it be "rainforests"? Generally, the phrasing is a little clusmy here as the groups found the in amber are not exclusively rainforest species (think of the camel spider or your arid-adapted hyids!). I think what you mean is something like "Among arachnids preserved in Burmese amber, all of their modern counterparts include species which can be found today in tropical rainforests".

References

Line 506: Journal style. Should it be "Fossiliferous Cretaceous amber from Myanmar (Burma): its rediscovery, biotic diversity, and paleontological significance" ?

Line 510: The Carsten Gröhn website "Ambertop" seems an slightly unusual choice of citation. I couldn't find the part of the website where he discusses the specific preservation methods referred to in the paper. Are there altenative references for the methodology such as Sadowski et al?
https://www.sciencedirect.com/science/article/pii/S0012825221001549

LIne 524: Style again "The phylogeny and classification of the Pseudoscorpionida (Chelicerata: Arachnida)." ?

Line 617: "The first Paleozoic Pseudoscorpions"

Figure legends

Line 683: should it be "Hyidae Chamberlin, 1930" (with comma) ?

Reviewer 2 ·

Basic reporting

The manuscript is clearly written in scientific language, all relevant and up to date literature references are included. The article structure is in accordance with standard practice, and the results are consistent with the data reported.

Experimental design

The investigation was rigorous and utilized all modern techniques for amber fossil remains.

Validity of the findings

The conclusions are justified by the data reported.

Additional comments

I have no further comments, The manuscript is extremely well presented and can be published with only minor editorial interventions.

---

## Round 0.2 · Minor Revisions

Thank you for referring to all reviewers' comments and correcting all the minor issues. Since the suggested corrections were minor, I assessed the revised version myself and I am happy with the current version. However, two minor issues remain to be solved to align with the journal requirement.

1. All the supplementary materials uploaded to public databases should be available at the latest upon publication. Additionally, a common practice to to make available not only the final reconstruction/model but also the raw image stack. Please, consider that.

2. If the amber piece was obtained before 2017, an ethical statement along the following lines should be mentioned: "The fossil was collected in full compliance with the laws of Myanmar and XX in 2015. To avoid any confusion and misunderstanding, all authors declare that to their knowledge, the fossil reported in this study was not involved in armed conflict and ethnic strife in Myanmar, and was acquired prior to 2017."

---

## Round 0.3 · accepted · Accept

Thank you for addressing all the comments. I am happy with the current version and the manuscript is ready for publication.